# The Effect of Co-Doping at the A-Site on the Structure and Oxide Ion Conductivity in (Ba_0.5−x_Sr_x_)La_0.5_InO_3−δ_: A Molecular Dynamics Study

**DOI:** 10.3390/ma12223739

**Published:** 2019-11-13

**Authors:** Kuk-Jin Hwang, Hae-Jin Hwang, Myung-Hyun Lee, Seong-Min Jeong, Tae Ho Shin

**Affiliations:** 1Energy & Environmental Division, Korea Institute of Ceramic Engineering and Technology, 101 Soho-ro, Jinju-si, Gyeongsangnam-do 52851, Korea; kjhwang@kicet.re.kr (K.-J.H.); mhlee@kicet.re.kr (M.-H.L.); 2School of Materials Science and Engineering, Inha University, 100 Inha-ro, Michuhol-gu, Incheon 22212, Korea; hjhwang@inha.ac.kr

**Keywords:** oxide ion conductivity, perovskite oxide, molecular dynamics simulation, ceramics electrolyte

## Abstract

A molecular dynamics simulation was used to investigate the structural and transport properties of a (Ba_0.5−x_Sr_x_)La_0.5_InO_3−δ_ (x = 0, 0.1, 0.2) oxygen ion conductor. Previous studies reported that the ionic conductivity of Ba-doped LaInO_3_ decreases because Ba dopant forms a narrow oxygen path in the lattice, which could hinder the diffusion of oxygen ions. In this study, we reveal the mechanism to improve ionic conductivity by Ba and Sr co-doping on an La site in LaInO_3_ perovskite oxide. The results show that the ionic conductivity of (Ba_0.5−x_Sr_x_)La_0.5_InO_3−δ_ increases with an increasing number of Sr ions because oxygen diffusion paths which contain Sr ions have a larger critical radius than those containing Ba ions. The radial distribution function (RDF) calculations show that the peak heights in compositions including Sr ions were lower and broadened, meaning that the oxygen ions moved easily into other oxygen sites.

## 1. Introduction

Ceramic ion conductors are important materials for application in electrochemical devices, such as solid oxide fuel cells (SOFCs), oxygen pumps, and oxygen sensors, due to their oxide ion conductivity and chemical and mechanical stability at high operating temperatures [1,2,3]. Oxide ion conduction occurs via a hopping mechanism through oxygen vacancies which is generated through the doping of low-valent cations to maintain the charge neutrality at the given composition. To achieve high ionic conductivity, oxygen ion conductors must have a large open space that allows a high level of point defect disorder and low migration enthalpy [4]. Some examples of such oxides are ZrO_2_, CeO_2_, Bi_2_O_3_ based oxide with fluorite structure, LaGaO_3_ based perovskites, Bi_4_V_2_O_11-_ and La_2_Mo_2_O_9_-based derivatives, Ba_2_In_2_O_5_ derived perovskite, and brownmillerite such as phases and pyrochlores [5].

Among them, fluorite-related structures such as yttria-stabilized zirconia (YSZ) show good performance as oxide ion conductors at high temperatures (~0.1 S·cm^−1^ at 1273 K). However, it is well known that the electrical conductivity of YSZ decreases to around 0.03 S·cm^−1^ at 1073 K [6]. Therefore, in most fluorite-related structures, a high temperature is required for efficient operation. To obtain a high ionic conductivity at intermediate temperatures (873–1073 K), perovskite and related oxides have been extensively studied. ABO_3_ perovskite oxides have a large number of oxygen vacancies and can be introduced into the lattice by the substitution of cation A or B with lower valence cations. Many researchers have studied lanthanum-gallate based ABO_3_ perovskite as an electrolyte for intermediate temperature SOFC (IT-SOFC) due to its high ionic conductivity and phase stability [7,8,9,10,11]. The perovskite oxide of La_0.8_Sr_0.2_Ga_0.8_Mg_0.2_O_3−δ_(LSGM) has a high oxide ion conductivity, which is higher than that of the YSZs [12]. Therefore, many types of research have been conducted to utilize these electrolytes in the SOFC. On the other hand, doped LaInO_3_ has rarely been introduced for electrolyte material: He et al. [13] studied the Sr-doped LaInO_3_ and Kim et al. [14] examined the Ba-doped LaInO_3_. However, these composites show a lower ionic conductivity than doped LaGaO_3_. To improve the ionic conductivity of LaInO_3_, Kakinuma et al. [15] reported that co-doped LaInO_3_ (La_0.5_Ba_0.3_Sr_0.2_)InO_2.75_ showed a very high conductivity, which was almost equal to that of La_0.8_Sr_0.2_Ga_0.8_Mg_0.2_O_3−δ_.

In previous studies, it was expected that the ionic conductivity of Ba-doped LaInO_3_ would increase with the formation of oxygen vacancies as the Ba ion was substituted [16]. However, a maximum level of conductivity was shown at a Ba content of 0.4–0.5 and conductivity by decreased because Ba ions act as a barrier for oxygen ion transport. Therefore, the aim of this study was to reveal the mechanism for improving ionic conductivity by Ba and Sr ions co-doping on an La site in LaInO_3_ using a structural investigation method and a molecular dynamics simulation.

## 2. Experimental and Simulation Methods

### 2.1. Experimental Method

The specimens of the (Ba_0.5−x_Sr_x_)La_0.5_InO_3−δ_ (x = 0–0.2) were prepared through the conventional solid-state reaction method. The starting reagents used in this study were La(NO_3_)_3_·6H_2_O (99%, Wako Chemical Co. Ltd., Osaka, Japan), BaO (90%, Acros, US), Sr(NO_3_)_2_/6H_2_O (99%, Sigma Aldrich, St. Louis, MO, USA) and In_2_O_3_ (99.99%, High Purity Chemicals Co. Ltd., Saitama, Japan). The precursors were mixed in a beaker with 100 mL of deionized water under magnetic stirring. This mixture was dried and calcined at 673 K for 2 h, followed by firing at 1773 K for 5 h for crystallization [14].

The crystalline phase of (Ba_0.5−x_Sr_x_)La_0.5_InO_3−δ_ (x = 0–0.2) was identified using X-ray diffractometer (D/max 2200V/PC, Rigaku, Akishima, Japan), with Cu Kα radiation (λ = 1.5406 Å) produced at 40 kV and 200 mA to scan the diffraction angles (2θ) between 10 and 90° with a step size of 0.02° at 2θ per second. The diffraction results were refined to analyze their crystal structure, inter-ionic distances and lattice parameters by the Rietveld refinement method using the GSAS EXPGUI software package (Los Alamos National Laboratory, Los Alamos, NM, USA) [17,18]. The crystallographic information file (CIF) for the cubic space group Pm-3m was taken from Uchimoto et al. [19] with lattice parameters a = b = c = 4.17214 Å.

For the electrical conductivity measurements, these composites were prepared into rectangular-shaped bars with a size of 3 mm × 3 mm × 10 mm. The platinum wires and platinum electrodes were formed by applying organic pastes (TR-7907, Tanaka Kinkinzoku Kogyo K. K., Tokyo, Japan) and firing at 1173 K to remove the organics. The conductivity was evaluated by using a general 4 terminal D.C. method and a custom jig with a Keithley 2400 Source Meter over a temperature range of from 600 to 1273 K with steps of 100 K at a rate of 5 °C·min^−1^. The obtained data were matched with the results in the previously reported study [15,20].

### 2.2. Simulation Method

The interatomic potentials employed in this study were from the Born model framework [21,22,23], consisting of a columbic term, a short-range repulsion term and a dispersion term as follows:(1)$$U_{ij}=\frac{q_{i}q_{j}}{r_{ij}}+f_{0}\left(b_{i}+b_{j}\right)exp\left[\frac{a_{i}+a_{j}-r_{ij}}{b_{i}-b_{j}}\right]-\frac{c_{i}c_{j}}{r_{ij}^{6}}$$
where *q_i_* and *q_j_* are the charges of two ions *i* and *j*, respectively. *r_ij_* is the distance between the *i* and *j* ions. *f*_0_ is a constant for unit adaptation (=1 kcal·mol^−1^·Å^−1^). The potential parameters *a*, *b* and *c* for each of the ions are shown in Table 1 [24,25]. The simulation models consist of 200 unit cells with a total of 950 atoms. Table 1 shows the three models characterized by their oxide ion conduction pathways by Sr contents. The pathway depends on the arrangement of atoms at the A-site positions and the three models with different oxide ion pathway values were obtained by manual arrangement of La, Sr and Ba ions.

All the calculations were carried out using the molecular dynamics simulation software LAMMPS (Sandia National Laboratory, Livermore, CA, USA) [26]. Newton’s equation of motion was integrated for 1600 ps with a time step of 1 fs after a relaxation step for 400 ps to remove the effect of the initial arrangement of oxygen vacancies. Each model with various compositions was simulated at temperatures from 873 K to 1273 K under a pressure of 1 bar. To derive the ionic conductivity, the mean square displacement (MSD) of oxygen ions was calculated according to
(2)$$MSD\left(t\right)=\frac{1}{N}\overset{N}{\underset{i=0}{\sum }}{\left(r\left(t\right)-r\left(0\right)\right)}^{2}$$
where *N* is the total number of ions and *r(t)* is the position of an ion *i* at the time t. MSD has a relation to the diffusion coefficient as the Einstein relation:(3)MSD(t)=6Dt
where *t* is the time and *D* is the diffusion coefficient. The ionic conductivity was obtained from the diffusion coefficient according to the Nernst-Einstein Equation:(4)$$\sigma =\frac{q^{2}ND}{fk_{B}TV}$$
where *σ* is the ionic conductivity, *k_B_* is the Boltzmann constant, *T* is temperature, *f* is the Haven ratio, *V* is volume and *q* is the ion’s charge. *f* had been calculated as 0.69 for the perovskite structure [27].

The radial distribution function (RDF) was calculated to analyze the interaction between oxide ions and cation pairs using the following equation:(5)$$g\left(r\right)=\frac{dN/N}{dV/V}=\frac{V}{N}\frac{N\left(r,\Delta r\right)}{4\pi r^{2}\Delta r}$$
where *V* is the simulation cell volume, *N* is the total number, *N*(*r*, Δ*r*) is the number of atoms found within a spherical shell of *r* to *r* + Δ*r* and the brackets represent a time average.

## 3. Results

The powder diffraction patterns of (Ba_0.5−x_Sr_x_)La_0.5_InO_3−δ_ (x = 0, 0.1, and 0.2) sintered at 1773 K are shown in Figure 1a. A single phase was observed in all the compositions and the X-ray diffraction patterns could be indexed as a simple cubic structure, indicating that 20 at.% of barium at the A-site could be substituted by strontium. It has been reported that LaInO_3_ exhibits an orthorhombic phase at room temperature and the cubic phase is formed by the doping of barium on the lanthanum site; as the doping content of barium increased in the La_1−x_Ba_x_InO_3−δ_ system, a mixture of cubic and orthorhombic phases was produced for the compositions of x = 0.1–0.3 and the single cubic phase was formed of the compositions of x = 0.4–0.8 [14]. According to a study on the lattice parameter in the La_1−x_Sr_x_InO_3−δ_ system, however, the solubility limit of strontium at the A-site in LaInO_3_ was reported to be about x = 0.1, although the ionic radius of strontium is close to that of lanthanum [13].Moreover, Ruiz-Trejo et al. [28] suggested that the best dopant was strontium on the lanthanum site by the calculation of the energy of alkaline-earth ions into the lanthanum site, indicating that the soluble amount of dopant at the A-site in LaInO_3_ may be independent to the ionic radius or the energy of the solution. It is interesting that the substituted amount of strontium was at least 20 at.% in the (Ba_0.5−x_Sr_x_)La_0.5_InO_3−δ_ system, whereas it was about 10 at.% in the La_1−x_Sr_x_InO_3−δ_ system. The existence of barium might enhance the substituted amount of strontium. Using these XRD results, the crystal structure was investigated using Rietveld refinement. Figure 1b shows the Rietveld refinement result of a (Ba_0.3_Sr_0.2_)La_0.5_InO_2.75_ composition as an example. The plus mark and the red line represent the experimental and calculated intensities, respectively; the blue line is the difference between them. The tsick marks (magenta) indicate the positions of Bragg peaks.

Figure 2 shows the lattice parameters of (Ba_0.5−x_Sr_x_)La_0.5_InO_3−δ_ as a function of Sr content. The lattice parameters decrease with an increasing Sr content because a Sr ion has a smaller ionic radius than a Ba ion. In addition, as the amount of Sr ions increases, the lattice constants decrease continuously, suggesting that Ba and Sr ions have been replaced by La ions and no secondary phase is formed. It is also indicated that oxygen vacancies are generated for charge compensation as Ba ions and Sr ions are substituted for La ion sites. The calculated lattice parameter is lower than the experimental values [20]. This is due to the incomplete nature of the interatomic potential parameters, as mentioned in the previous study [29]. In this study, the calculated lattice parameters were ~0.3% lower than the experimental values, indicating that the interatomic potentials used in this study were reasonably accurate.

Figure 3a,b show the ternary oxides with an ABO_3_ cubic perovskite structure and the oxide ion pathway of the oxygen octahedron site in the perovskite structure, which consists of four A-site ions and two B-site ions. In the perovskite structure, oxide ions can be moved through the oxygen pathway and the radius of the inscribed circle of the oxygen pathway is called the critical radius, as shown in Figure 3b, so we investigated the diffusion behavior of all oxide ions. Figure 3c shows the calculated mean square displacement (MSD) of each ion in Sr contents at 1073 K. One of the main purposes of MSD analysis is the extraction of the diffusion coefficient value from the simulation [30]. In Figure 3c, the MSDs for all the ions in the (Ba_0.3_Sr_0.2_)La_0.5_InO_3−δ_ composite are presented at 1073 K. The figure clearly shows that the MSD of oxygen ions continuously increases with time. However, the MSD of cations has a constant value and there is no cation diffusion in the perovskite oxides. These results show that (Ba_0.5−x_Sr_x_)La_0.5_InO_3−δ_ is a pure oxide ion conductor [15]. In Figure 3d, we focused on the MSD of oxygen ions as the Sr ion content increased, which clearly demonstrates that the oxygen ion transport property is increased as the amount of Sr ions increases.

Figure 4 shows an Arrhenius plot of the electrical conductivity with experimental values at the temperature range of 873 K to 1273 K. In this study, we considered (Ba_0.5−x_Sr_x_)La_0.5_InO_3−δ_ as an oxide ion conductor and the electron conduction behavior was ignored. Kakinuma et al. [15] reported that the (Ba_0.5−x_Sr_x_)La_0.5_InO_3−δ_ is a pure oxide ion conductor and Kim. et al. investigated the electrical conduction behavior of Ba-doped LaInO_3_ and reported that this composition has pure oxide ion conduction under a dry N_2_ atmosphere [14]. In Figure 4, the experimental values are about 3% lower than in the previous study [20]. There is a small difference, but it tends to be similar to the results of Kakinuma et al., which is a reliable result. The calculated ionic conductivity for (Ba_0.5−x_Sr_x_)La_0.5_InO_3−δ_ was lower than the experimental data for all compositions. The interatomic potential used for this research is the empirical potential. It is difficult to exactly reproduce the experimental values due to the limitations, so there is a small error between the calculated and experimental values. The ionic conductivity, σ, is represented by three terms, i.e., carrier concentration, C, carrier charge, Ze, where Z is the valence and e is the electronic charge, and carrier mobility, μ.
(6)σ=CZeμ

In this study, all the compositions have the same number of carriers, i.e., oxygen ions, but a higher ionic conductivity was obtained in higher contents of Sr ions. However, the activation energy increased slightly with an increasing number of Sr ions. In order to investigate the effect of Sr ion substitution on the ionic conductivity and carrier-mobility increase, the tolerance factor and lattice free volume were considered.

For the understanding of the oxygen ion conduction in the perovskite-structured oxides, several parameters, such as tolerance factor, lattice free volume, and critical radius, influenced the oxygen ion conductivity [31,32,33]. Figure 5 shows the relationship between ionic conductivity, lattice free volume and tolerance factor depending on the amount of Sr ions. The lattice free volume was defined as the difference between the unit cell volume and the summed volume occupied by all constituent ions. As the free volume increases, the lattice spacing increases and the carrier ions can be moved easily, therefore, ionic conductivity increases. Figure 5 shows that the lattice free volume of (Ba_0.5−x_Sr_x_)La_0.5_InO_3−δ_ decreased with an increasing Sr content and ionic conductivity increased. In addition, the Goldschmidt tolerance factor in the following equation decreased with an increase in Sr ions.
(7)$$G_{t}=(r_{A}+r_{O})/\left(\sqrt{2}\left(r_{B}+r_{O}\right)\right)$$
where, r_A_, r_B_ and r_O_ are the ionic radii of A cations, B cations and oxygen ions, respectively. The Goldschmidt tolerance factor can predict the structural stability depending on the degree of distortion of the BO_6_ octahedron in the perovskite structure. The ideal cubic structure has a value of 1 without the tilt of BO_6,_ and the further the value deviates from 1, the lower the structural symmetry and the lower the conductivity too. Inaguma et al. [34] reported that in perovskite-structured oxides with the same B-ion, the B–O distances are independent of A-site ions, although the lattice parameter and free volume decrease with decreases in the ionic radii of the A-site ions. This means that the BO_6_ octahedron tilts to maintain the same ionic distance between B and O ions as Sr ions, having a small ionic radius, are replaced with Ba ion sites. As a result of tilting, the activation energy increased, as shown in Figure 4. Therefore, we expected that the size of the oxygen-ion pathway and electrical conductivity would be decreased. However, in this study, the opposite result was obtained. Although the tolerance factor was decreased and BO_6_ tilting occurred, the effect on electrical conductivity is considered to be insignificant in our research. Therefore, other factors affect the electrical conductivity in (Ba_0.5−x_Sr_x_)La_0.5_InO_3−δ_ compositions.

Table 2 shows the critical radius of oxygen ion pathways with different Sr ion contents in (Ba_0.5−x_Sr_x_)La_0.5_InO_3−δ_. The critical radius was defined as the critical size of the triangle formed by two A-site cations and one B-site cation where oxygen ions move into adjacent oxygen vacancies as shown in Figure 3b. As the critical radius is large, the ionic conductivity has a high value. As shown in Table 2, the critical radius is larger in the pathway with Sr ions rather than with Ba ions. As the amount of Sr ions increased the critical radius also increased and the tolerance factor and lattice free volume decreased. When La ion sites are substituted with low-valent cations with a large ionic radius, the ionic conductivity depends on the critical radius among the various factors in LaInO_3_ perovskite. These results will be useful for predicting the ionic conduction properties when the synthesis of perovskite oxides is performed.

Figure 6 shows the probability of oxygen ion transport according to the oxygen ion pathway. Nm/Np represents the amount of movement of the oxygen ions per adjacent oxygen ion pair by breaking the A–O ionic bonds. The binding energies of O–Sr and O–Ba were −1664 and −1469 kJ/mol, respectively, and more energy would be required to break an O–Sr bond compared with an O–Ba bond. Therefore, oxide ions can be easily moved through the oxygen pathway, including Ba ions. However, in all compositions, the number of oxygen ions moving through the △LaLaIn pathway occupies the greatest proportion. The ratio of passage through △LaBaIn was smaller than that of △LaSrIn, and it was confirmed that the critical radius is a crucial factor in revealing the oxygen transport property in a doped LaInO_3_ perovskite structure. 

To discover the structural difference between the Sr doped and the undoped compositions, the interionic distances between particular ions and oxygen ions were analyzed using the radial distribution function (RDF). Figure 7a–c shows the RDF curves for the ionic pairs of La–O, In–O and O–O in (Ba_0.5−x_Sr_x_)La_0.5_InO_3−δ_. In all the compositions, there were no significant differences, however, as the Sr ions are substituted, the peak of RDF is lower and broadens, which suggests that a doped composition is a distorted form derived from the undoped composition [35,36,37]. Figure 7c indicates that the oxygen ion does not exist in the original position and the substitution of Sr ions to A sites makes it easy to move the oxygen ions because it permits the amount of oxygen vacancies in neighboring O ions to be estimated. Because the higher peak intensity of the O–O pair is indicative of a smaller number of oxygen-vacancy pairs in the lattice structure, the undoped composite is estimated to have fewer oxygen-vacancy pairs than the Sr-doped composite. This means that more conduction paths are available in the Sr-doped composite, thus, that a higher ionic conductivity would be expected. This study suggests that to achieve a high ionic conductivity in doped LaInO_3_, the critical radius should be = larger by the substitution of cations.

## 4. Conclusions

In this study, the oxygen ion conductivity and the conduction mechanisms of Ba- and Sr-doped LaInO_3_ were analyzed using a molecular dynamics simulation. The (Ba_0.5−x_Sr_x_)La_0.5_InO_3−δ_ compositions are verified as oxygen-

ion conductors by calculation of the MSD. As the Sr ion content increased, the MSD of oxygen ions and ionic conductivity increased too. The substitution ofSr ions with small ionic radii at the Ba ions sitereduced the tolerance factor and lattice free volume but the critical radius increased. Therefore, the critical radius is a dominant factor in doped LaInO_3_ perovskite. As a result of the RDF calculations, the oxygen ions tend to deviate from the original oxygen ion site in doped Sr ion composition. To achieve high ionic conductivity, the oxygen pathway should have a large critical radius in doped LaInO_3_.

## Figures and Tables

**Figure 1 materials-12-03739-f001:**
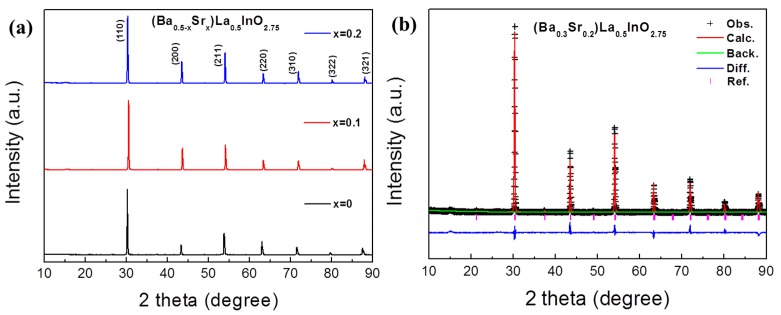
(**a**) X-ray diffraction spectra of (Ba_0.5−x_Sr_x_)La_0.5_InO_3−δ_ (x = 0, 0.1, and 0.2) sintered at 1773 K for 5 h. (**b**) Rietveld refinement of the (Ba_0.3_Sr_0.2_)La_0.5_InO_3−δ_.

**Figure 2 materials-12-03739-f002:**
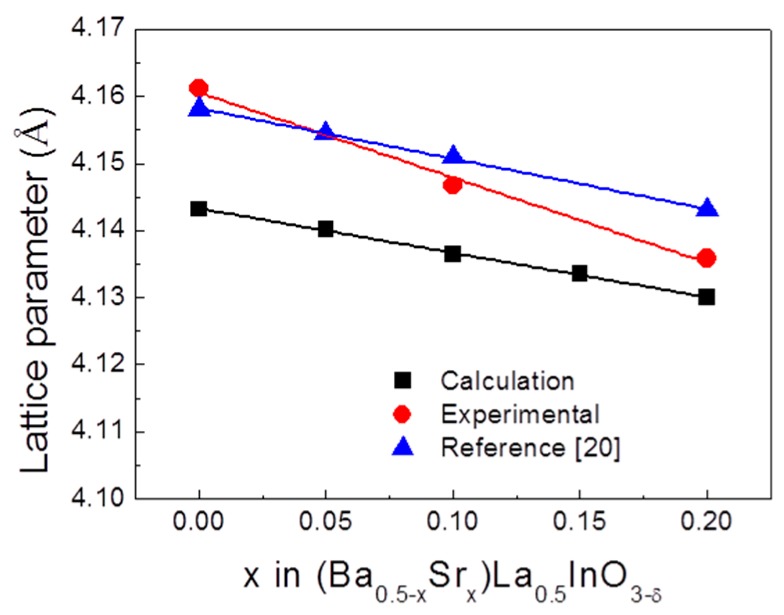
Lattice parameters of (Ba_0.5−x_Sr_x_)La_0.5_InO_3−δ_ as a function of Sr content compared with experimental and reference values [20].

**Figure 3 materials-12-03739-f003:**
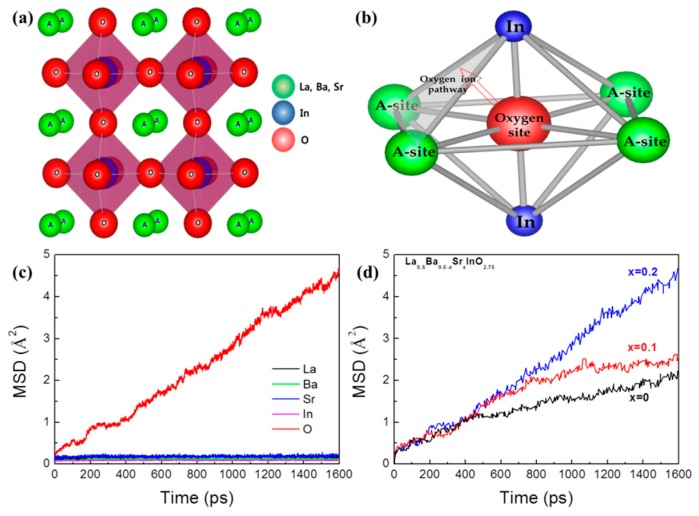
(**a**) Typical ABO_3_ perovskite structure, (**b**) Schematic diagram of the oxygen octahedral site and the oxygen pathway, (**c**) Mean square displacement (MSD) of each ion in (Ba_0.3_Sr_0.2_)La_0.5_InO_3−δ_, (**d**) MSD of oxygen ions in Sr contents at 1073 K.

**Figure 4 materials-12-03739-f004:**
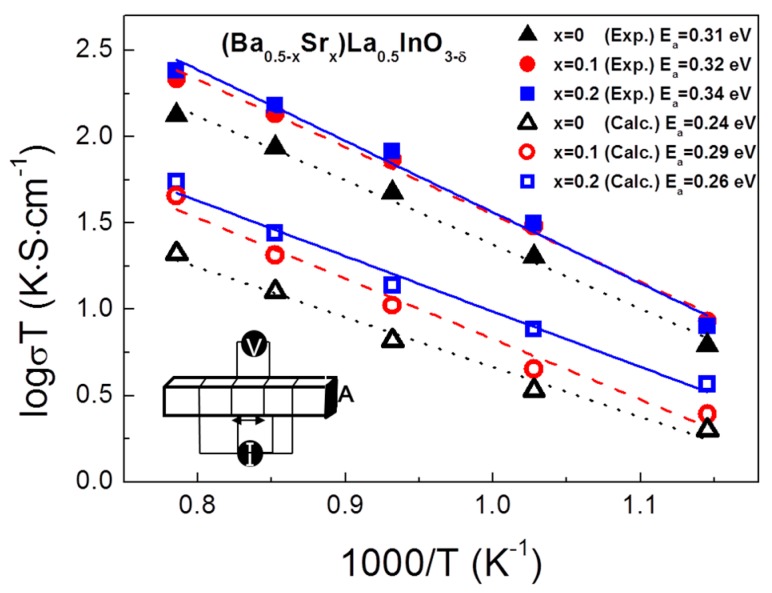
Arrhenius plots for the electrical conductivity of (Ba_0.5−x_Sr_x_)La_0.5_InO_3−δ_ at 873–1273 K.

**Figure 5 materials-12-03739-f005:**
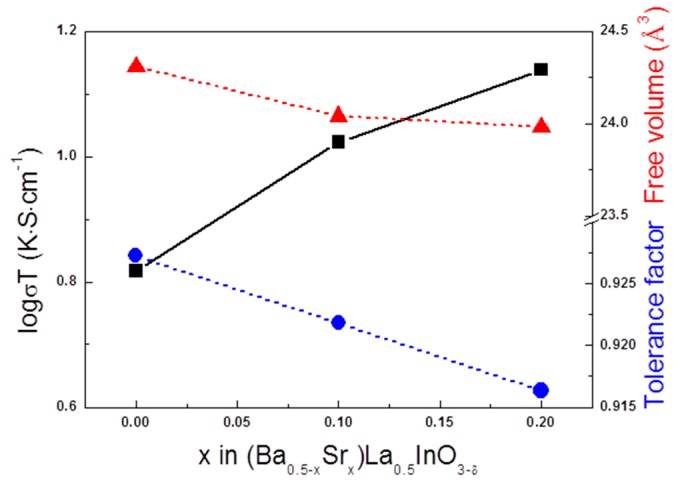
Ionic conductivity and tolerance factor (blue), free volume (red) of (Ba_0.5−x_Sr_x_)La_0.5_InO_3−δ_ at 1073 K.

**Figure 6 materials-12-03739-f006:**
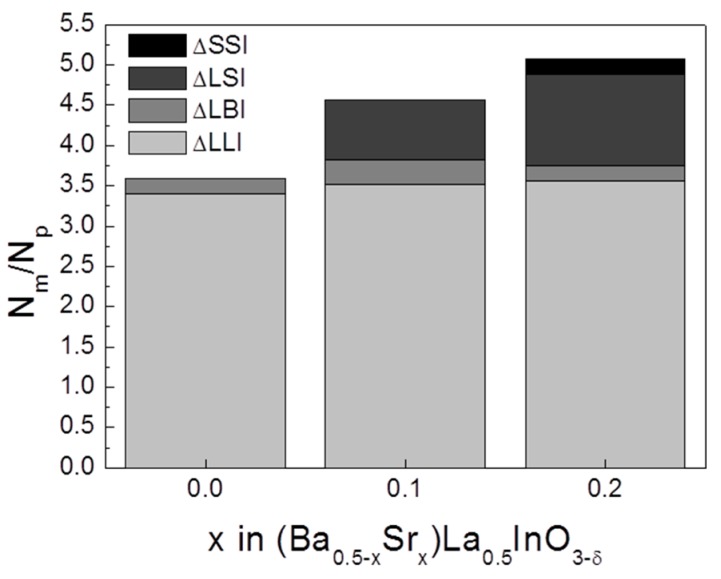
Probability of oxygen ion migration through the pathway in (Ba_0.5−x_Sr_x_)La_0.5_InO_3−δ_ at 1073 K.

**Figure 7 materials-12-03739-f007:**
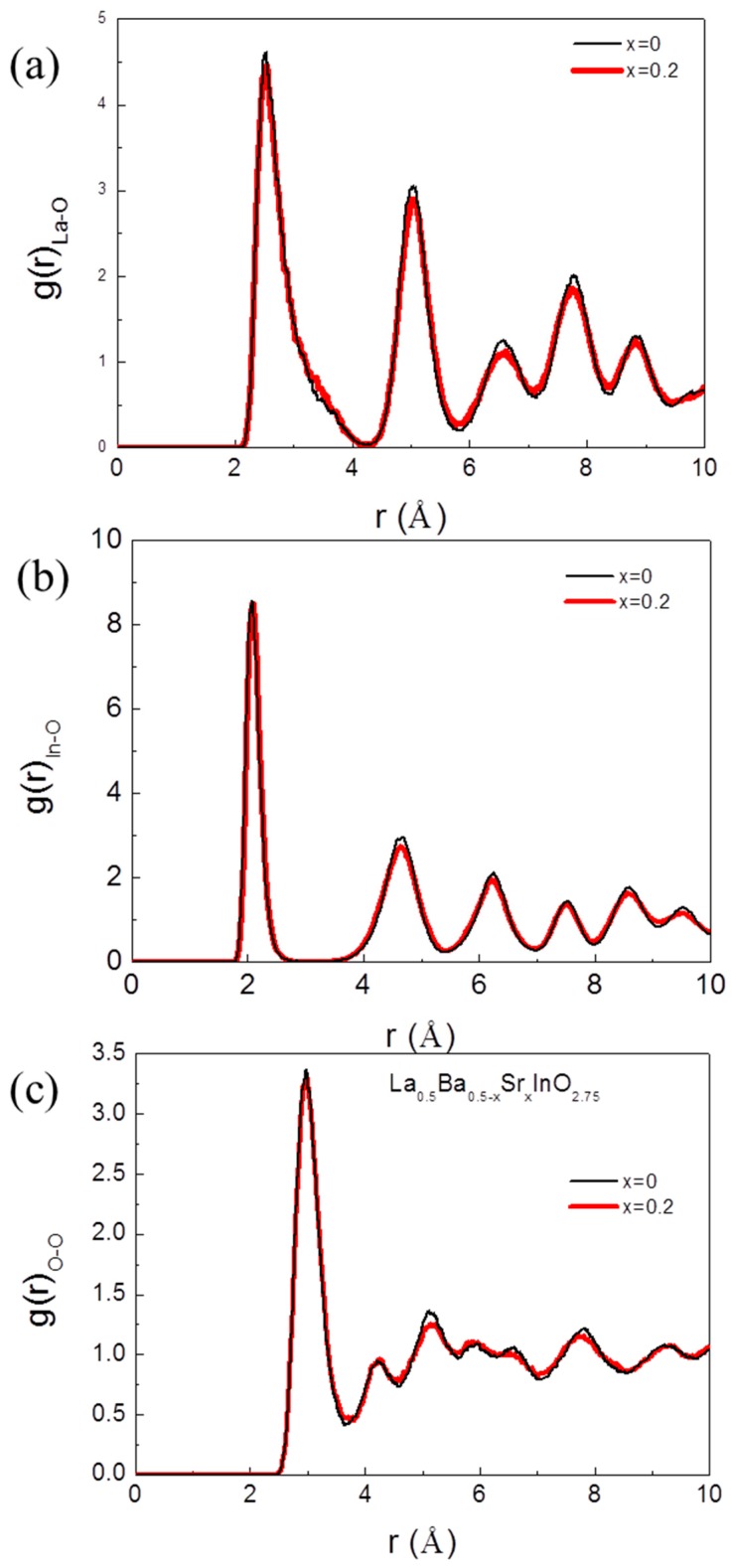
RDF (**a**) La O ions, (**b**) In–O ions and (**c**) O O ions in (Ba_0.5−x_Sr_x_)La_0.5_InO_3−δ_ at 1073 K.

**Table 1 materials-12-03739-t001:** Interatomic potential parameters and configurations of oxide ion conduction pathways with different Sr ion contents for (Ba_0.5−x_Sr_x_)La_0.5_InO_3−δ_.

Interatomic Potential Parameters						
-		La	Ba	Sr	In	O
a (Å)		2.149	1.800	2.028	2.013	1.568
b (Å)		0.205	0.077	0.194	0.220	0.087
c (kcal·Å^6^/mol)^0.5^		0	0	0	0	27
**Configurations of Oxide Ion Conduction Pathways**						
x	△LaLaIn	△LaBaIn	△LaSrIn	△BaSrIn	△SrSrIn	△BaBaIn
0	572	1256				572
0.1	572	964	292	172	8	392
0.2	572	716	540	284	68	220

**Table 2 materials-12-03739-t002:** The critical radii of oxygen ion pathways consist of two A-site ions and one B-site ion with different Sr ion contents in (Ba_0.5−x_Sr_x_)La_0.5_InO_3−δ_.

-	△LaLaIn	△LaBaIn	△BaBaIn	△LaSrIn	△SrSrIn	△BaSrIn
x = 0	0.993	0.925	0.832	-	-	-
x = 0.1	0.998	0.918	0.829	0.962	0.967	0.874
x = 0.2	0.996	0.906	0.832	0.954	0.95	0.883

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
