# Peer review of "The Effect of Co-Doping at the A-Site on the Structure and Oxide Ion Conductivity in (Ba0.5−xSrx)La0.5InO3−δ: A Molecular Dynamics Study"

_materials, 2019, doi:10.3390/ma12223739_

Round 1

Reviewer 1 Report

This paper studies the oxygen ion conductivity and conduction mechanism of Ba and Sr doped LaInO3. It is reasonable and interesting. However, from reviewer’s view, some revisions can be made as follows:

It would be better to provide the figure of experimental apparatus. Besides, the instrumental error is not illustrated in this paper during the measurement of electrical conductivity, please add this part in Experimental Method. The Eqs. (1) and (5) are not clear enough, please improve the clarity. In line 104, ‘B’ should be written in the subscript form in Boltzmann constant kB. The caption of Fig. 1(b) could be added in the text and the title could be more concise. In Figure 2, the values of x in calculation, experiment, and reference are not consistent, why? Why are the experimental values higher than the calculated results in Figure 4? Please add more error analysis in the paper. It would be better to add a Nomenclature part.

Author Response

Reviewer 1

General Comments: This paper studies the oxygen ion conductivity and conduction mechanism of Ba and Sr doped LaInO3. It is reasonable and interesting. However, from reviewer’s view, some revisions can be made as follows:

Comment 1: It would be better to provide the figure of experimental apparatus. Besides, the instrumental error is not illustrated in this paper during the measurement of electrical conductivity, please add this part in Experimental Method.

Response to C1: First of all, thank you for your worthy comments. The electrical conductivity measurement method is briefly added to Figure 4, and the results of this experiment showed a difference of about 3% compared to the previous results and showed a similar tendency. Therefore, our results are reliable. The manuscript was revised according to the reviewer’s comment as following:

The obtained data was matched with the results in a previously reported study of Kakinuma et al. [20]. In this study, it was about 3% lower than the previous study [20]. There is a small difference, but it tends to be similar to the results of Kakinuma et al., which is a reliable result.

Comment 2:  The Eqs. (1) and (5) are not clear enough, please improve the clarity.

Response to C2: The equations were revised according to the reviewer’s comment.

Comment 3:  In line 104, ‘B’ should be written in the subscript form in Boltzmann constant kB.

Response to C3: The Boltzmann constant was revised according to the reviewer’s comment “kB”.

Comment 4:  The caption of Fig. 1(b) could be added in the text and the title could be more concise.

Response to C4: Thank you for your comments and we reflect your worthy comments in the revised manuscript and figure caption according to the reviewer’s comment as following:

Figure 1(b) shows the Rietveld refinement result of a (Ba0.3Sr0.2)La0.5InO2.75 composition as an example. The plus mark and red line represent the experimental and calculated intensities, respectively; the blue line is the difference between them. Tick marks (magenta) indicate the positions of Bragg peaks. Figure 1. (a) XRD spectra of (Ba0.5-xSrx)La0.5InO3-δ (x=0, 0.1, and 0.2) sintered at 1773 K for 5 h. (b) Rietveld refinement of the (Ba0.3Sr0.2)La0.5InO3-δ.

Comment 5:  In Figure 2, the values of x in calculation, experiment, and reference are not consistent, why? Why are the experimental values higher than the calculated results in Figure 4? Please add more error analysis in the paper.

Response to C5: We fully agree with your reasonable comments. This study employed Born Mayer potential, which was known as the best one to predict oxide ion conduction behaviour in the perovskite. However, some calculations with this potential reported lower calculated conductivity than with experimental data in the oxide conductor [29]. Hence, we estimate that the reason for the low conductivity relies on the imperfection of the employed interatomic potential. Of course, the development of perfect interatomic potential capable to calculate precisely might be best solution. However, in this study, our best efforts were not in the precise calculation of the ionic conductivity, but in the introduction of governing rule of ionic conductivity in the perovskite oxide. The manuscript was revised considering the reviewer’s comment as following:

The calculated lattice parameter is lower than the experimental values. It is due to the incomplete of the interatomic potential parameters, as mentioned in the previous study [29]. In this study, the calculated lattice parameters were ~0.3 % lower than the experimental values indicating that the interatomic potentials used in this study were reasonably accurate. 29. Burbano, M.; Norberg, S.T.; Hull, S.; Eriksson, S.G.; Marrocchelli, D.; Madden, P.A.; Watson, G.W.; Oxygen vacancy ordering and the conductivity maximum in Y2O3-doped CeO2. Chem. Mater. 2012, 24, 222-229. https://doi.org/10.1021/cm2031152.

Comment 6:  It would be better to add a Nomenclature part.

Response to C6: The abbreviations such as RDF and MSD were re-defined at the beginning of every section to clarify the meaning.

Reviewer 2 Report

Research presented by Hwang et al. discusses Ba and Sr doped LaInO3 perovskite materials as high temperature oxygen ion conductors. The primary aim of the research is to combine experimental methods and the simulations to show the role of Ba and Sr ions in oxygen transfer efficacy. Authors discuss their results in the light of the rather well-known La0.8Sr0.2Ga0.8Mg0.2O3-δ perovskites. The idea behind the research is primarily based on the previously reported (Ba0.3Sr0.2La0.5)InO2.75 (Solid State Ion. 2004, 175, 139-14) material which exhibited conductive performance comparable to La0.8Sr0.2Ga0.8Mg0.2O3-δ.

In general, I found the manuscript well written with adequate amount of information provided to make the rationale of the research clear to the reader. I do not have any objection regarding the quality of research presented in this manuscript. However, for the further clarity of the text, I believe that some minor corrections or improvements are needed:

As I understood, the B-side ions in Figure 3b are In ions. The B-sides must be clearly marked on the figure for clarity. In Line 148-149, it is stated that “in the perovskite structure, oxide ion can be moved through the oxygen pathway as shown in Figure 3(b), so we investigated the diffusion behavior of all oxide ions”. Why is this pathway preferred by oxygen transport? Is it directly related to the “critical radius of oxygen ion pathway with different Sr ion contents” explained in Table 2? If so, a connection between the statement in line 148-149 must be clearly made. Could you please provide additional information regarding this point? In Table 2, please replace ΔLLI, ΔLBI… with ΔLaLaIn, ΔLaBaIn… as in Table 1. I personally had difficulties to understand what ΔLLI, ΔLBI… stand for. In line 185-192, authors are reporting a deviation from Goldschmidt tolerance factor and reverse relation between ionic conductivity. They relate the deviation to the absence of BO6 distortion. Could Authors suggest an explanation how the absence of distortion is causing deviation from Goldschmidt tolerance factor or reverse relation with ionic conductivity? I think, a discussion on this issue in Results section or a comment in Conclusions section should be provided to enrich the impact of research.

Author Response

Reviewer 2

General Comments: Research presented by Hwang et al. discusses Ba and Sr doped LaInO3 perovskite materials as high temperature oxygen ion conductors. The primary aim of the research is to combine experimental methods and the simulations to show the role of Ba and Sr ions in oxygen transfer efficacy. Authors discuss their results in the light of the rather well-known La0.8Sr0.2Ga0.8Mg0.2O3-δ perovskites. The idea behind the research is primarily based on the previously reported (Ba0.3Sr0.2La0.5)InO2.75 (Solid State Ion. 2004, 175, 139-14) material which exhibited conductive performance comparable to La0.8Sr0.2Ga0.8Mg0.2O3-δ.

In general, I found the manuscript well written with adequate amount of information provided to make the rationale of the research clear to the reader. I do not have any objection regarding the quality of research presented in this manuscript. However, for the further clarity of the text, I believe that some minor corrections or improvements are needed:

Comment 1:  As I understood, the B-side ions in Figure 3b are In ions. The B-sides must be clearly marked on the figure for clarity.

Response to C1: The figure was revised according to the reviewer’s comment and the manuscript was also revised to clarify Figure 3 as follows:

Figure 3(a) and (b) show the ternary oxides with ABO3 cubic perovskite structure and oxide ion (red circle) pathway of oxygen octahedron site in the perovskite structure, which is consisted of four A-site ions (green circle) and two B-site ions (blue circle).

Comment 2:  In Line 148-149, it is stated that “in the perovskite structure, oxide ion can be moved through the oxygen pathway as shown in Figure 3(b), so we investigated the diffusion behaviour of all oxide ions”. Why is this pathway preferred by oxygen transport? Is it directly related to the “critical radius of oxygen ion pathway with different Sr ion contents” explained in Table 2? If so, a connection between the statement in line 148-149 must be clearly made. Could you please provide additional information regarding this point?

Response to C2: We appreciate your comments and fully agree with your idea. The radius of the inscribed circle of the oxygen ion pathway in Figure 3(b) is the critical radius, and the manuscript has been modified for clarity according to the reviewer’s comment as following:

Figure 3(a) and (b) show the ternary oxides with ABO3 cubic perovskite structure and oxide ion pathway of oxygen octahedron site in the perovskite structure, which is consisted of four A-site ions and two B-site ions. In the perovskite structure, oxide ion can be moved through the oxygen pathway, and the radius of the inscribed circle of the oxygen pathway is called the critical radius, as shown in Figure 3(b), so we investigated the diffusion behaviour of all oxide ions.  

Comment 3:  In Table 2, please replace ΔLLI, ΔLBI… with ΔLaLaIn, ΔLaBaIn… as in Table 1. I personally had difficulties to understand what ΔLLI, ΔLBI… stand for.

Response to C3: The table was revised according to the reviewer’s comment.

Comment 4:  In line 185-192, authors are reporting a deviation from Goldschmidt tolerance factor and reverse relation between ionic conductivity. They relate the deviation to the absence of BO6 distortion. Could Authors suggest an explanation how the absence of distortion is causing deviation from Goldschmidt tolerance factor or reverse relation with ionic conductivity? I think, a discussion on this issue in Results section or a comment in Conclusions section should be provided to enrich the impact of research.

Response to C4: Thank you for your worthy comments. Thus we revised as your comments. The ideal cubic structure has a 1 without the tilt of BO6 and the further the value deviates from 1, the lower the structural symmetry and the lower the conductivity too. Inaguma et al. [34] reported that in perovskite-structured oxides with the same B-ion, the B-O distances are close independent of A-site ions, though the lattice parameter and free volume decrease with decreases in ionic radius of A-site ion. This means that BO6 octahedron tilts to maintain the same ionic distance between B and O ions as Sr ions having a small ionic radius are replaced with Ba ion sites. As a result of tilting, the size of oxygen ion pathway and electrical conductivity was expected to decrease. However, in this study, the opposite result was obtained. Although tolerance factor is decreased and BO6 tilting occurred, the effect on electrical conductivity is considered to be insignificant in our research. Therefore, other factors affect the electrical conductivity in (Ba0.5-xSrx)La0.5InO3-δ compositions.

Inaguma, Y.; Katsumata, T.; Mori, D. Predominant factor of activation energy for ionic conductivity in perovskite-type lithium ion-conducting oxides. J. Phys. Soc. Jpn. 2010, 79, 69-71.

Reviewer 3 Report

The authors present a study investigating the trends in ionic conductivity of LaInO3, as a function of metal ion doping. Experimental data as well as trends identified from molecular dynamics simulations are used to propose a mechanism explaining the nature of ionic conduction pathways. Authors have identified some interesting trends from such an analysis. However, the article reads incoherently in several sections arising largely from issues in grammar and sentence structure. As such, the work is of relevance to researchers in the field of SOFCs, metal oxides etc. and may be appropriate for publication in this journal. Further recommendations are listed below:                                                                                             

Specific Comments/Questions:                  

Authors describe a solid-state synthesis for the metal oxides in lines 61-66. Please explain any optimization protocols or reasoning behind these chosen conditions. Alternatively, if this synthetic protocol was directly adapted from prior studies, please provide references! Please do not use abbreviations such as ‘RDF’ in the abstract. Furthermore, abbreviations such as RDF and MSD should be re-defined once at the beginning of every section to aid the reader. Authors describe the Arrhenius plot in figure 4 as a plot of ionic conductivity vs. temperature. However, the measurement of conductivity in the experimental details section (lines 74-79) was done via 4-point probe measurements – this relates to electrical conductivity and not ionic conductivity. It is unclear how electrical and ionic conductivity are related herein. Figure 2 describes the trends in lattice parameters as a function of Sr doping in the metal oxide – again where is this lattice parameter derived from? No details are provided to this effect. Table 2 describes the changes in critical radius of the oxygen ion in different pathways/configurations labeled with three letters e.g. SSI. Similar abbreviations are used in the discussion in lines 210-218. Please clarify to the reader that each refers to a pathway fully described in Table 1 e.g. SSI refers to SrSrIn. 

Author Response

Reviewer 3

General Comments: The authors present a study investigating the trends in ionic conductivity of LaInO3, as a function of metal ion doping. Experimental data as well as trends identified from molecular dynamics simulations are used to propose a mechanism explaining the nature of ionic conduction pathways. Authors have identified some interesting trends from such an analysis. However, the article reads incoherently in several sections arising largely from issues in grammar and sentence structure. As such, the work is of relevance to researchers in the field of SOFCs, metal oxides etc. and may be appropriate for publication in this journal. Further recommendations are listed below:                                                                                            

Comment 1: Authors describe a solid-state synthesis for the metal oxides in lines 61-66. Please explain any optimization protocols or reasoning behind these chosen conditions. Alternatively, if this synthetic protocol was directly adapted from prior studies, please provide references!

Response to C1: Thank you very much for your comments, we added a reference according to the reviewer’s comment.

Comment 2: Please do not use abbreviations such as ‘RDF’ in the abstract.

Response to C2: We fully agree with your comments; Thus the manuscript was revised according to the reviewer’s comment.

Comment 3: Furthermore, abbreviations such as RDF and MSD should be re-defined once at the beginning of every section to aid the reader.

Response to C3: The manuscript was revised according to the reviewer’s comment.

Comment 4: Authors describe the Arrhenius plot in figure 4 as a plot of ionic conductivity vs. temperature. However, the measurement of conductivity in the experimental details section (lines 74-79) was done via 4-point probe measurements? this relates to electrical conductivity and not ionic conductivity. It is unclear how electrical and ionic conductivity are related herein.

Response to C4: The manuscript was changed from ionic conductivity to electronic conductivity according to the reviewer’s comment. Kakinuma et al. [15] reported that the (Ba0.5-xSrx)La0.5InO3-δ is the pure oxide ion conductor and Kim. et al. investigated the electrical conduction behaviour of Ba-doped LaInO3 and reported that this composition has pure oxide ion conduction under a dry N2 atmosphere [14]. Therefore, we considered (Ba0.5-xSrx)La0.5InO3-δ is oxide ion conductor and electron conduction behaviour was ignored.

Comment 5: Figure 2 describes the trends in lattice parameters as a function of Sr doping in the metal oxide ? again where is this lattice parameter derived from? No details are provided to this effect.

Response to C5: We added a reference according to the reviewer’s comment.

20. Kakinuma, K.; Yamamura, H.; Atake, T. High oxide ion conductivity of (Ba1-x-ySrxLay)InO2.5+y/2 members derived from Ba2In2O5 system. Defect. Diffus. Forum. 2005, 242-244, 159-168.

Comment 6: Table 2 describes the changes in critical radius of the oxygen ion in different pathways/configurations labeled with three letters e.g. SSI. Similar abbreviations are used in the discussion in lines 210-218. Please clarify to the reader that each refers to a pathway fully described in Table 1 e.g. SSI refers to SrSrIn.

Response to C6: The manuscript was revised according to the reviewer’s comment.

Reviewer 4 Report

The work by Kuk-Jin Hwang et al. reports a molecular dynamic study in relationship with crystal structure and electrical conductivity of  Ba,Sr-doped LaInO3. The authors clearly define the research problem, they clearly show the importance of this study and the results would be worthy for publication in the Materials. The manuscript presents the new interesting information,  indeed, the free volume often does not explain the conductivity behavior, and such an approach as the use of the critical radius can be effectively used and may be extended to other systems.

The comments:

The authors do not discuss the dependence of the activation energy of conductivity on the strontium content. If the critical radius changes, should the activation energy change? A discussion on this should be added.

The authors in Table 1 use the notation as LaLaIn (and so on), but in Table 2 the designations are different LLI (and so on), authors need to make the same notation. Or authors should make the Table  2 and legend more comprehensible, the dimension of the radius must be specified.

Line 65: The authors write “This solution …”. Why solution? Indium oxide is insoluble in water. The authors need to use a different term or explain if they dissolved indium oxide in HNO3.

Line 104: It should be “kB” - subscript character

Line 114: It should be “LaInO3” - subscript character

Line 120: It should be “strontium is close” not “ similar.”

Authors need to indicate the publication number in the inset to Fig. 2

Line 160: It should be “ , ”  before (d)

Line 241: It should be “compositions”  not  “ composites.”

Recommendations for the future, not for corrections.

1) The authors use hydrates for synthesis, however, these substances are not a weight form. TG measurements are required to confirm the exact amount of crystallization water.

2) Measurements of the DC conductivity should be replaced by AC conductivity, since the contribution of the polarization resistance can be significant. In addition, to discuss bulk conductivity (which is associated with structural features) it is necessary to exclude the grain boundary response.

3) The discussion should relate to ionic conductivity, therefore proof of the absence of an electronic contribution should be made (measurement of transport numbers).

Author Response

Reviewer 4

General Comments: The work by Kuk-Jin Hwang et al. reports a molecular dynamic study in relationship with crystal structure and electrical conductivity of Ba,Sr-doped LaInO3. The authors clearly define the research problem, they clearly show the importance of this study and the results would be worthy for publication in the Materials. The manuscript presents the new interesting information, indeed, the free volume often does not explain the conductivity behavior, and such an approach as the use of the critical radius can be effectively used and may be extended to other systems.

Comment 1: The authors do not discuss the dependence of the activation energy of conductivity on the strontium content. If the critical radius changes, should the activation energy change? A discussion on this should be added.

Response to C1: The activation energy increased with increasing Sr ion, but the difference is very small from 0.31 to 0.34 eV in experimental results and from 0.24 to 0.26 eV in simulation results. As the Sr ion increases, the tolerance factor decreases, which is related to the distortion of BO6. In this study, however, the critical radius was found to have a greater effect on oxygen ion conduction. In other words, the combination of A-site and B-site ions is an important factor in this composition having a cubic structure than the activation energy.

However, the activation energy increased slightly with increasing Sr ions. As a result of tilting, the activation energy increased as shown in Figure 4.

Comment 2: The authors in Table 1 use the notation as LaLaIn (and so on), but in Table 2 the designations are different LLI (and so on), authors need to make the same notation. Or authors should make Table 2 and legend more comprehensible, the dimension of the radius must be specified.

Response to C2: The manuscript was revised according to the reviewer’s comment.

Comment 3: Line 65: The authors write “This solution …”. Why solution? Indium oxide is insoluble in water. The authors need to use a different term or explain if they dissolved indium oxide in HNO3.

Response to C3: The manuscript was revised according to the reviewer’s comment as following:

This mixture was dried and calcined at 673 K for 2 h, following by firing at 1773 K for 5 h for crystallization

Comment 4: Line 104: It should be “kB” - subscript character

Comment 5: Line 114: It should be “LaInO3” - subscript character

Comment 6: Line 120: It should be “strontium is close” not “ similar.”

Comment 7: Authors need to indicate the publication number in the inset to Fig. 2

Comment 8: Line 160: It should be “ , ” before (d)

Comment 9: Line 241: It should be “compositions” not  “ composites.”

Response to C4-C9: The manuscript was revised according to the reviewer’s comment.

Recommendations for the future, not for corrections.

1) The authors use hydrates for synthesis, however, these substances are not a weight form. TG measurements are required to confirm the exact amount of crystallization water.

2) Measurements of the DC conductivity should be replaced by AC conductivity, since the contribution of the polarization resistance can be significant. In addition, to discuss bulk conductivity (which is associated with structural features) it is necessary to exclude the grain boundary response.

3) The discussion should relate to ionic conductivity, therefore proof of the absence of an electronic contribution should be made (measurement of transport numbers).

Response to Recommendations: Thank you very much for considering our manuscript. In future work, we will conduct our research through your recommendations.

Round 2

Reviewer 3 Report

In the response to review comments, authors have clarified the interplay between ionic and electronic conductivity herein.

i.e."Kakinuma et al. [15] reported that the (Ba0.5-xSrx)La0.5InO3-δ is the pure oxide ion conductor and Kim. et al. investigated the electrical conduction behaviour of Ba-doped LaInO3 and reported that this composition has pure oxide ion conduction under a dry N2 atmosphere [14]. Therefore, we considered (Ba0.5-xSrx)La0.5InO3-δ is oxide ion conductor and electron conduction behaviour was ignored."

Please include this justification in the manuscript as well - to clarify this relationship to the reader. 

Author Response

Comments: In the response to review comments, authors have clarified the interplay between ionic and electronic conductivity herein.

i.e."Kakinuma et al. [15] reported that the (Ba0.5-xSrx)La0.5InO3-δ is the pure oxide ion conductor and Kim. et al. investigated the electrical conduction behaviour of Ba-doped LaInO3 and reported that this composition has pure oxide ion conduction under a dry N2 atmosphere [14]. Therefore, we considered (Ba0.5-xSrx)La0.5InO3-δ is oxide ion conductor and electron conduction behaviour was ignored."

Please include this justification in the manuscript as well - to clarify this relationship to the reader. 

Response: Thank you very much for considering our manuscript. The manuscript was revised according to the reviewer’s comment in line 168 as following:

In this study, we considered (Ba0.5-xSrx)La0.5InO3-δ is oxide ion conductor and electron conduction behaviour was ignored. Kakinuma et al. [15] reported that the (Ba0.5-xSrx)La0.5InO3-δ is the pure oxide ion conductor and Kim. et al. investigated the electrical conduction behaviour of Ba-doped LaInO3 and reported that this composition has pure oxide ion conduction under a dry N2 atmosphere [14].

Reviewer 4 Report

The manuscript can be accepted in the present form.

Author Response

Comments: The manuscript can be accepted in the present form.

Response: Thank you very much for considering our manuscript.